# Ternary regulation mechanism of *Rhizoma drynariae* total flavonoids on induced membrane formation and bone remodeling in Masquelet technique

**Ding Li[1], Dun Zhao[1], Zhikui Zeng[2], Feng Huang[1], Ziwei Jiang[1], Hao Xiong[1], Tianan Guan[1], Bin Fang[1], Yue Li**[1] *

1 Department of Orthopedic, The First Affiliated Hospital of Guangzhou University of Chinese Medicine, Guangzhou, China, 2 Department of Orthopedics, The Affiliated Hospital of Jiangxi University of Chinese Medicine, Nanchang, China

* doctor_liyue@vip.126.com

## Abstract

### Context

*Rhizoma drynariae* total flavonoids (RDTF) are used to treat fractures. CD31^hiEmcn^hi vessels induced by PDGF-BB secreted by osteoclast precursors, together with osteoblasts and osteoclasts, constitute the ternary regulatory mechanism of bone tissue reconstruction.

### Objective

This study aimed to determine whether RDTF can promote bone tissue remodeling and induce membrane growth in the rat Masquelet model and to explore its molecular mechanism based on the ternary regulation theory.

### Methods

Thirty-six SD rats were randomized to three groups: blank, induced membrane, and RDTF treatment (n = 12/group). The gross morphological characteristics of the new bone tissue were observed after 6 weeks. Sixty SD rats were also randomized to five groups: blank, induction membrane, low-dose RDTF, medium-dose RDTF, and high-dose RDTF (n = 12/group). After 4 weeks, immunohistochemistry and western blot were used to detect the expression of membrane tissue-related proteins. The mRNA expression of key factors of ternary regulation was analyzed by qRT-PCR.

### Results

RDTF positively affected angiogenesis and bone tissue reconstruction in the bone defect area. RDTF could upregulate the expression of key factors (PDGF-BB, CD31, and endomucin), VEGF, and HMGB1 mRNA and proteins in the ternary regulation pathway.

**Data Availability Statement:** All relevant data are within the paper and its Supporting Information files.

**Funding:** This study was funded by National Natural Science Foundation of China [No.81603640, No.81974575] and Natural Science Foundation of Guangdong Province [No. 20171206].The funders had no role in study design, data collection and analysis, decision to publish, or preparation of the manuscript.

**Competing interests:** All authors declare that there is no conflict interest.

## Discussion and conclusion

Although the expected CD31$^{hi}$Emcn$^{hi}$ vessels in the induction membrane were not observed, this study confirmed that RDTF could promote the secretion of angiogenic factors in the induced membrane. The specific mechanisms still need to be further studied.

## 1 Introduction

Bone defects caused by severe trauma, infection, tumor resection, and other reasons are common in clinical practice, which seriously affects patients' quality of life and causes significant economic and social burden. The treatment of bone defect is a challenge for the surgeons, numerous options are use in the clinical such as free cancellous bone grafts, allogeneic bone, vascularized fibula graft and distraction osteogenesis. But these techniques have their limitations, like bone resorption, long period of treatment, delayed union or nonunion. The free cancellous bone grafts and vascularized fibula graft are often used to repair small size bone defects, but not for critical bone defects (defects >6 cm), [1]. Allogeneic bone and artificial bone are highly dependent upon vascularization, use the traditional surgery technique, it is difficult to achieve a sufficient blood supply [2, 3]. Distraction osteogenesis is an efficient technique to treat the critical bone defects, but the long period of treatment made the patients suffering a painful process.[4–7]. How to improve the efficiency and shorten the period of treatment is a challenge for the orthopedist.

In 1986, Masquelet proposed the concept of "induced membrane" to prevent the invasion of adjacent soft tissues into the bone defect, maintain the autograft in place, preserve the autograft volume and producing osteoinductive substances locally [8–10]. The Masquelet technique is a two-stage procedure, the first stage involved debridement of bone and soft tissues, insertion of a polymethylmethacrylate (PMMA) cement spacer, bone stabilization.After 4 to 8 weeks, a membrane surrounding the spacer was formed and then the second stage surgery was operated to remove the spacer for implantation of bone grafts. A series of reports have showed that Masquelet technique can be used on the reconstruction of bone defect. However, it also has disadvantages such as poor osteogenesis quality of new bone, and infection [10–12], joint contracture, bone nonunion, and stress fracture [13]. The formation of the induced membrane and its ability to induce bone remodeling is the key point of the Masquelet technique. The mesenchymal stem cells (MSCs) from the induced membrane and normal bone marrow show the same osteogenic potential [14]. The membrane are rich in osteoblasts, osteoclasts, osteoclast precursor cells, secrete vascular endothelial growth factor (VEGF), transforming growth factor (TGF-β1), bone morphogenetic protein-2 (BMP-2), and other growth factors that promote bone formation [10, 15]. CD31$^{hi}$Emcn$^{hi}$ vessels play an important role in bone reconstruction [16]. During the process of bone formation and resorption, the decrease of CD31$^{hi}$Emcn$^{hi}$ angiogenesis results in a significant reduction in the number of trabecular and cortical bones [16]. CD31$^{hi}$Emcn$^{hi}$ angiogenesis is primarily regulated by platelet-derived growth factor-BB (PDGF-BB). PDGF-BB can induce CD31$^{hi}$Emcn$^{hi}$ angiogenesis by promoting endothelial progenitor cells (EPCs) and MSCs. For this reason, CD31$^{hi}$Emcn$^{hi}$ vessels, along with osteoclasts and osteoblasts, constitute the "ternary regulation" mechanism of bone remodeling.

*Rhizoma drynariae* (Chinese name, Gusuibu) is a traditional Chinese herb with a variety of pharmacological activities [17, 18] and is used for the prevention or treatment of bone-related diseases [19, 20]. The major bioactive constituents of are flavonoids, with potential roles in

osteoporosis prevention and the treatment of disc degeneration [21, 22]. *Rhizoma drynariae* total flavonoids (RDTF) have some efficacy for improving proliferation and differentiation [23, 24], inflammation [25], fracture healing, and teeth osteocyte protection [26]. But, whether the RDTF can improve the efficiency of Masquelet technique is still unknown.

In this study we focused on whether RDTF can promote induce membrane growth and bone remodeling in the rat Masquelet model and explore its molecular mechanism based on ternary regulation theory.

## 2 Material and methods

### 2.1 Animals and Masquelet's technique model establishment

Masquelet's technique was performed in a rat femur defect model. Totally ninety-six specific pathogen-free (SPF) male rats (8–10 weeks of age, weighing 260–280 g) were provided by the Laboratory Animal Center of Guangzhou University of Traditional Chinese Medicine(ethical number: SCX-2018-0034). All animals were treated in compliance with the principles and procedures contained in the most recent publication of the National Institutes of Health guide for the care and use of laboratory animals. The rats were housed four animals/cage, at 24–26°C and 60–80% humidity, with 14 h of light/day. They received rat chow and water ad libitum. For the stage one, rats were general intraperitoneal anesthesia (5% chloral hydrate, 8 mg/kg), the right hindlimb was shaved, and the animals were placed in a lateral position. Penicillin (80,000 U, North China Pharmaceutical Co. Ltd, China) was injected intramuscularly before surgery to prevention of infection. A longitudinal incision was made in the skin and superficial fascia over the right femur. The biceps femoris and vastus lateralis muscles were separated and elevated from the greater trochanter, exposing the lateral aspect of the femoral bone. A six-hole, 1.5-mm homemade stainless steel mini-plate was applied to the lateral aspect of the femur shaft. Two 1.5-mm cortical screws (Suzhou Yuelai Medical Equipment Co., Ltd., Jiangsu, China) were drilled in holes #1 and #6 for fixation (S1 Fig). Then, two Kirshner wires were drilled into holes #2 and #5. The osteotomy guide plate was placed above the steel plate after passing through the holes. With the osteotomy guide plate was secured, a critical-sized defect (CSD) measuring 4 mm was created in the femur bone shaft, using a reciprocating saw. The bone defect was filled with PMMA cement and molded into a cylindrical shape. After securing the cement with silk thread, the wound was washed with sterile water, and the fascia and skin were sewn. 4 weeks after the stage one surgery the stage 2 surgery was conducted, preoperative preparation and surgical approach were the same as that of the first stage. The induced membrane was carefully cut open, PMMA cement was removed, and two segments of rat coccyx were taken as bone graft materials to implant in the bone defect area. The induced membrane tissue was sutured, and muscle, deep fascia, and skin were sutured in turn. The animals were returned to their cages and house separated. Penicillin was given for 5 days postoperatively, during which time the rats were monitored daily for the occurrence of abnormal behavior and/or complications. Meloxicam (0.2mg/kg) in the first 3 days after operative to analgesia. Under general anesthesia, the membranes surrounding PMMA cement spacer were harvested 4 weeks after the first stage. After the experiment, the animals were euthanized with an overdose (150 mg/kg) of pentobarbital administered intraperitoneally.

### 2.2 Treatment

Experiment 1, 60 rats were randomly divided into five groups (empty control group, EG; induced membrane group, IG; low dose RDTF group, LG; medium dose group, MG; high dose group, HG). The operation for the bone defect was performed as above, and no second stage surgery was performed. The EG group was not implanted with PMMA cement.

Approximately 24 h after the animal model was established, the LG, MG, and HG groups were administered with QiangGu capsules (contain 180mg RDTF/each, National Medicine Permit No. Z20030007; batch #180304) intragastrically. Mixed the powder in distilled water, and administered by gavage, the intervention concentrations were 157.5 mg/kg/d for HG, 78.75 mg/kg/d for MG, and 39.38 mg/kg/d for LG. The animals in the EG and IG groups were given the same amount of pure water. Gavage was started on the first day after modeling and was performed once a day until the sampling day of the fourth week after surgery. Histological and radiographs were performed routinely at 4 weeks postoperatively. The rats with complications were not included in the analyses to avoid confounding factors from infections and inflammation.

Experiment 2, 36 rats were randomly divided into three groups (empty control group, ECG; induced membrane group, IMG; induced membrane plus total flavonoids group, IFG). The operation process of the bone defect was performed as above. The ECG group was not implanted with PMMA cement. The second stage surgery was performed 4 weeks after PMMA cement implantation. Bone grafting was not performed in the ECG group, and the postoperative treatment was the same as that in the first stage. QiangGu capsules (contain 180mg RDTF/each, National Medicine Permit No. Z20030007; batch #180304) were used. Approximately 24 hours after the animal model was established, the IMG and IFG groups were administered QiangGu capsules mixed in distilled water., the dose was 78.75 mg/kg/d, The ECG and IMG groups were fed the same dose of 0.9% normal saline intragastrically every day for six weeks. All 36 rats tolerated the operation and were kept in single cages during the experiment.

## 2.3 Micro CT and angiography analyses

After general anesthesia with an intraperitoneal injection of 5% (w/v) chloral hydrate (0.8 ml/100 g), anteroposterior and lateral radiographs of the right hindlimb were obtained using X-ray (50 kV and 4 mA; Kodak, Rochester, NY, USA) at 4 weeks after the operation. For the ECG, IMG, and IFG, the radiographs were obtained 6 weeks after the second-stage surgery. The radiographs were scored using the Lane-Sandhu scoring system. Scoring was carried out by three experts in the department of radiology.

For angiography, after anesthesia, the skin and subcutaneous tissues were incised under the xiphoid process of the sternum. The heart was exposed by an incision along the manubrium sternum and ribs. The perfusion needle was inserted from the right atrium to reach the aortic arch. The left atrial appendage was cut open, and 1% heparin saline was perfused until the fluid was clear and there was no blood outflow. After paraldehyde infusion and fixation, the Microfil (Microfil, MV-122, Flow Tech, Inc.) contrast agent was injected until it was completely drained (nearly 20 ml), put the animal in 4˚C overnight, and then collected the femur, decalcification thoroughly. Use the Micro-CT to scan and analysis.

For the Micro-CT scan, femur were collected, and the attached soft tissue was removed thoroughly and then fixed in 4% paraformaldehyde. The specimens were scanned with an animal micro-CT scanner (Skyscan 1172, Bruker, Madison, WI, USA). The Scanning parameters with an 80 kVp energy setting, intensity of 313 uA with 280 ms acquisition time and a image resolution of 120 μm. The defective region was identified by a contour as a traced region of interest (ROI).

## 2.4 Histoological analysis

Histological evaluations of the induced membranes were performed by microscope observations of stained tissue sections. Cryosections (5-μm thick) of the induced membrane were

stained with hematoxylin and eosin (H&E) (Solaibao Technology Co., Ltd., Beijing, China). The components of the extracellular membrane were detected with an Elastica-van Gieson stain (Merck Serono KGaA, Darmstadt, Germany) and Muscovy O-solid green cartilage staining (Feijing Biotechnology Co., Ltd., Beijing, China), according to the manufacturers' instructions.

## 2.5 Quantitative real-time polymerase chain reaction (qRT-PCR)

Pieces of membranes were immediately frozen after collection and stored in liquid nitrogen. Frozen membrane (200–600 mg) was powdered with a pestle and homogenized in TRIzol (Promega, Madison, WI, USA). RNA was used for cDNA synthesis with the RevertAid First-Strand Synthesis Kit (Thermo Fisher Scientific, Waltham, MA, USA) using oligo-dT as a primer. cDNA product (1 μl) was used for qRT-PCR with the SYBER Green technique (SYBR Green/ROX qPCR Master Mix) by quantifying the product at each cycle in a real-time manner. The amount of product was monitored for 40 cycles. The product specificity was confirmed by melting curve and, in some cases, by visualizing the products with agarose gel electrophoresis. The expression of β-actin was studied using the primer 5'-GAT CAA GAT CAT TGC TCC TCC TG-3' and 5'-AGG GTG TAA AAC GCA GCT CA-3' (annealing temperature of 60˚C). The primers were 5'-CAC CCA AGT GTT TTG GCA CC-3' and 5'-ATG AGC CCT TTC TTC CAC GG-3' for CD31, 5'-GTG CAA CAC CCA AAA CAC CT-3' and 5'-TGA ACT CTG ATT CTC CGT CTT GT-3' for Endomucin, 5'-GTC GGA GAG CAA CGT CAC TA-3' and 5'-TGC GCT TTC GTT TTT GAC CC-3' for VEGF, 5'-ACT TGA ACA TGA CCC GAG CA-3' and 5'-GAA CAC CTC TGT ACG CGT CT-3' for PDGF-BB, and 5'-ACC GCC CAA AAA TCA AAG GC-3' and 5'-ATA GGG CTG CTT GTC ATC CG-3' for HMGB1, all with an annealing temperature of 66˚C. GADPH was used as endogenous controls, relative expression level was computed using 2-ΔΔCt method.

## 2.6 Western-blot

The induced membrane proteins were collected by lysis in RIPA buffer, a BCA protein assay was used to measure the protein content. After centrifugation for 10 min, the extracted protein was put in boiling water for 10 min. The proteins were isolated by 10% polyacrylamide gel electrophoresis at 120 mV, and transferred to PVDF membranes at 200 mA for approximately 1h. Subsequently, the membranes were blocked using 5% non-fat dried milk and 4% BSA for 2 h at 25˚C, followed by incubation with the following primary antibodies: anti-GAPDH (1:1000, Shanghai Kangcheng Co., China), anti- HMGB1 (1:1000, ab79823, Abcam, Cambridge, United Kingdom), anti-VEGFA (1:1000, ab46154, Abcam, Cambridge, United Kingdom), anti-CD31 (1:1000, ab182981, Abcam, Cambridge, United Kingdom), and anti-PDGF-BB (1:1000, ab16829, Abcam, Cambridge, United Kingdom), anti-Endomucin (1:1000, sc-65495, Santa Cruz Biotechnology, Santa Cruz, CA, USA). Subsequently, the membranes were incubated with the appropriate horseradish peroxidase (HRP)-conjugated secondary antibody (1:10,000, ZB-2301, Wuhan Bode Biological Engineering Co, Ltd, Wuhan, China) for 1h on a shaker at 37˚C. Finally, after ECL chemiluminescence, development, and fixation, the film was scanned. Sensiansys software in a JS-680A automatic gel imaging analysis system was used for the grayscale analysis of the target bands.

## 2.7 Immunohistochemistry

Sections were obtained as mentioned above, and the slides were incubated overnight. The slides were washed for 15 min in 0.1% Triton X-100 diluted with PBS and blocked with 5%

bovine serum albumin (ZSGB-BIO, Beijing, China) for 30 min. The slides were incubated in the following antibodies overnight at 4˚C: CD31 anti-collagen I (1:2000, ab182981, Abcam, Cambridge, United Kingdom), endomucin anti-collagen I (1:50, sc-65495, Santa Cruz Biotechnology, Santa Cruz, CA, USA), PDGF-BB anti-collagen I (1:100, ab16829, Abcam, Cambridge, United Kingdom), VEGFA anti-collagen I (1:100, ab46154, Abcam, Cambridge, United Kingdom), HMGB1 anti-collagen I (1:100, ab79823, Abcam, Cambridge, United Kingdom). The slides were washed with PBS AND incubated with a fluorescent secondary antibody (AlexaFluor®488) for 50 min at 37˚C. DAB was used to stain the nuclei. The slides were observed under a fluorescence microscope (Motic, Wetzlar, Germany). ImageJ software (National Institutes of Health, Bethesda, MD, USA) was used to images the results at 200× magnification.

## 2.8 Statistical analysis

SPSS 22.0 was used for statistical analysis. The continuous data were expressed as mean ± standard deviation (SD) and were analyzed using Student's t-test (two-group comparisons) or ANOVA with the LSD post hoc test (multiple group comparison). Categorical data were presented as n (%) and analyzed using the chi-square test. P-values <0.05 were considered statistically significant.

# 3 Results

## 3.1 Examination of the bone defects and the Masquelet technique model

X-ray at 4 weeks after surgery showed that the alignment of the osteotomy fracture in the observation group was good. The plates and screws were in place, and the stable 4-mm bone defect area was maintained (Fig 1A). For the PMMA implantation, the cement was in a normal position without loosening or overturning (Fig 1B).

The induced membranes showed many small and round cells near the bone cement. A rich vascular network was formed far from the bone cement and adjacent to the muscle, while fibrous tissue parallel to the PMMA bone cement block was formed in the middle area (Fig 1C).

## 3.2 RDTF improves ternary regulation related factors

Except for the EG group, the other groups had a more typical inductive membrane structure and more abundant vascular distribution (Fig 2A). To further explore the potential machanism of RDTF in Induced membrane formation, some related biomarkers were detected from protein and gene level respectively.

The immunohistochemistry results demonstrated that the expression of the CD31 protein in the EG group was weaker than that in the other groups (Fig 2B). The endomucin and PDGF-BB proteins were expressed in the induced membrane tissues of each group, but all of them were weakly positive. HMGB1 was expressed abundantly in the induced membrane tissue of each group.

PCR results showed that compared with the empty control group, the mRNA expression of CD31, VEGFA, HMGB1, and PDGF-BB in the IG, LG, MG, and HG were significantly increased. The expression of endomucin mRNA was not significantly increased, and the expression level was significantly increased only when with high-dose RDTF. After the use of RDTF, the ternary regulation-related factors VEGFA and HMGB1 tended to increase. After the use of RDTF, regardless of the dose, the expression of PDGF-BB mRNA was significantly increased compared with the IG group, while the expression of HMGB1 and endomucin

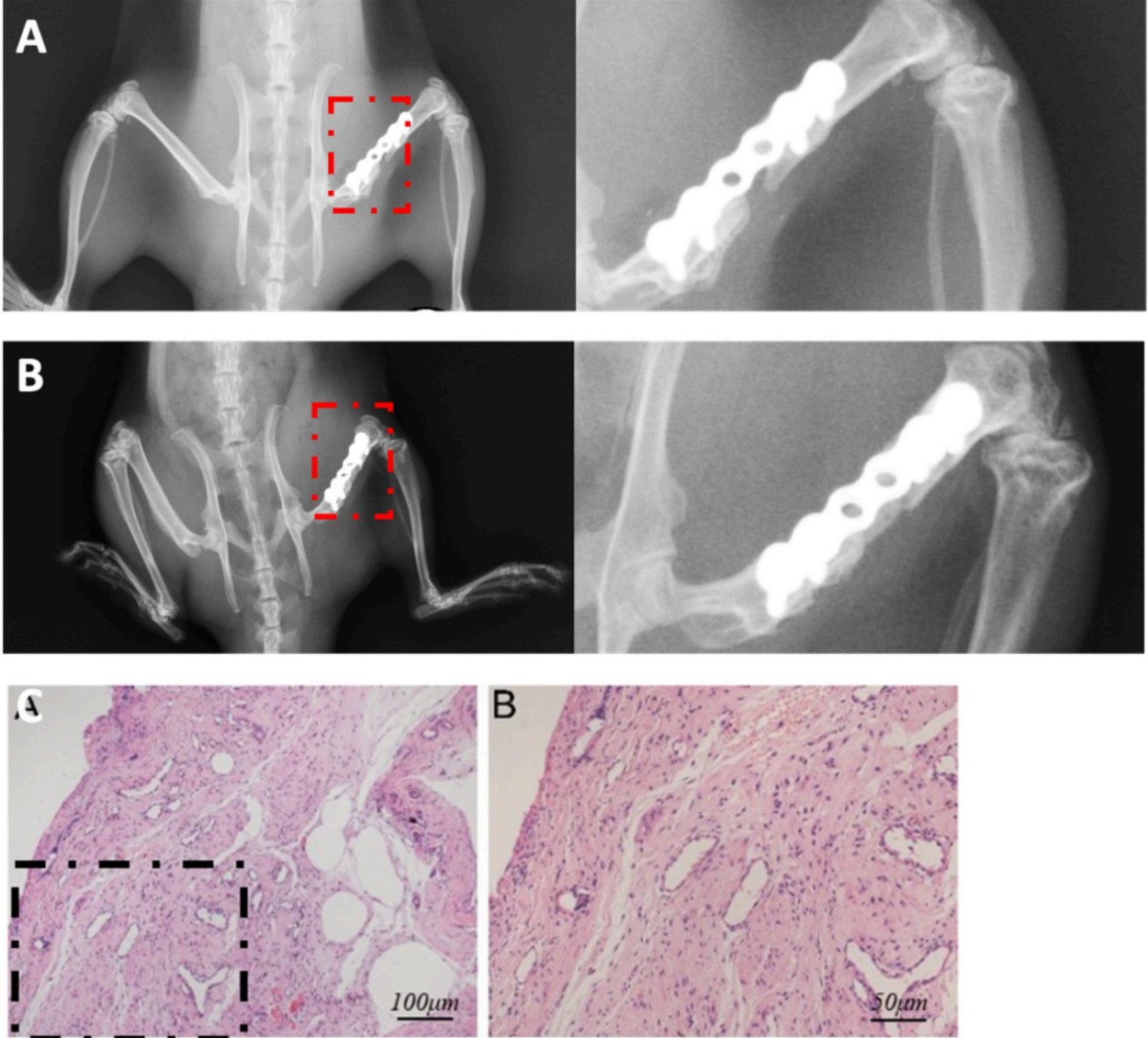

**Fig 1. Characterization of the bone defects** (A) X-ray of the typical models. X-ray showed good limb strength line, stable 4-mm bone defect, and no loosening of the screws and plates. (B) Hematoxylin & eosin staining comparing the membranes harvested from bone defects (control group) and beside the PMMA cement (model group).

significantly increased only when high doses of RDTF were used compared with those of IG. And VEGFA expression was significantly increased when the dose of RDTF reached the medium dose. The results of CD31 also showed a general trend of increased expression (Fig 3A). Similar results were observed at the protein level (Fig 3B).

### 3.3 RDTF improves bone quality and blood vessels in the bone defect area

Six weeks after surgery, as presented in Fig 4, PMMA was radiopaque so that the bone substitute and the broken ends were visible. The defect in the IFG group showed continuous osteogenesis. Cortical bone was well reconstructed, and the marrow cavity had formed. The defect in the IMG group showed discontinuous osteogenesis. The bone marrow cavity was not fully formed, and the reconstructed bone cortex was weak, with discontinuous bone cortex still

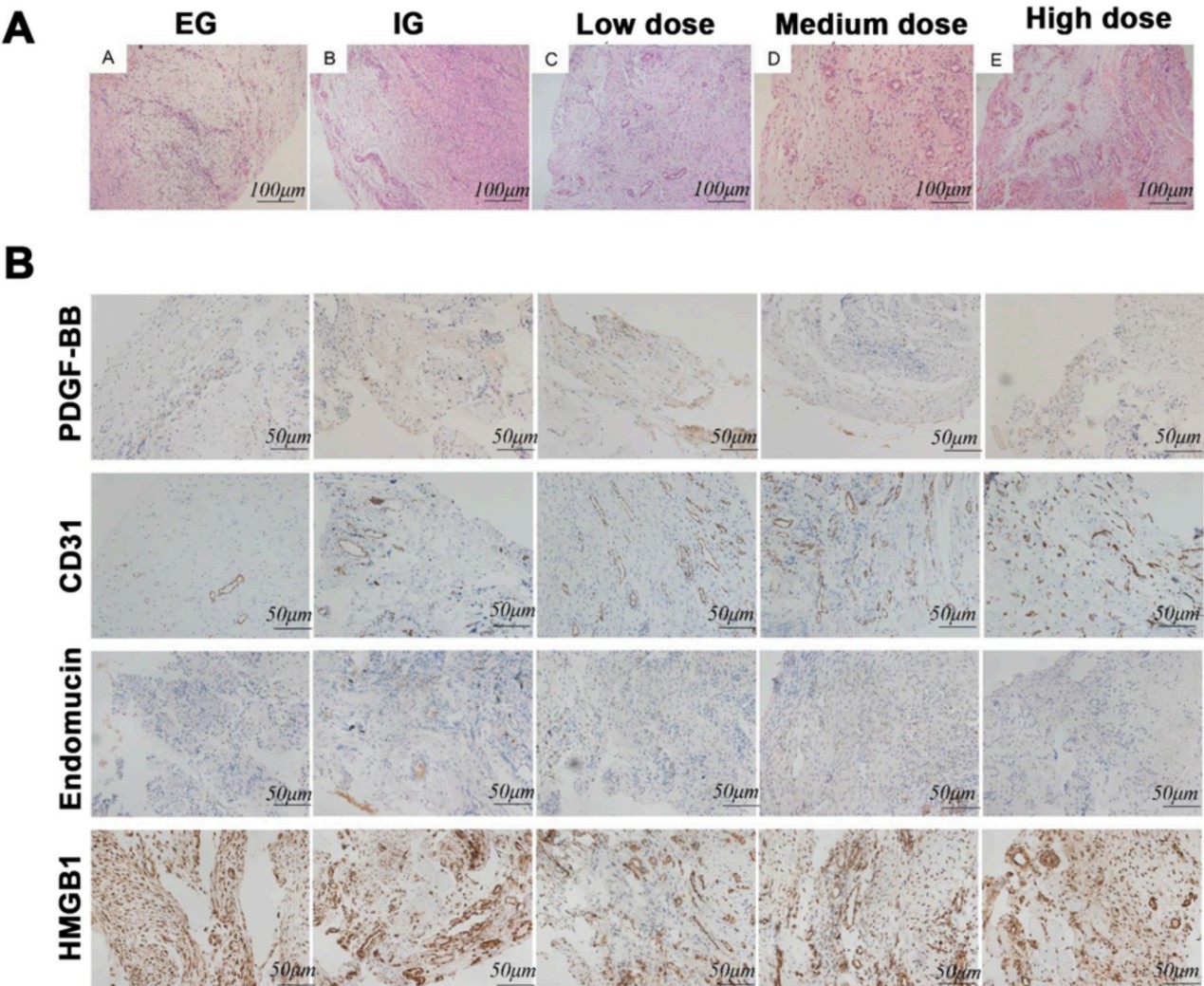

**Fig 2. Histological examination and immunohistochemistry in the five groups (empty control group, EG; induced membrane group, IG; low dose RDTF group, LD; medium dose group, MG; high dose group, HG).** (A) Hematoxylin and eosin staining of the induced membrane in each group. (B) Immunohistochemistry for PDGF-BB, CD31, endomucin, and HMGB1.

visible. Nothing grew in the critical-sized bone defects in the ECG group. The micro-CT cross-sections showed more obvious new bone formation in the IFG group than in the other two groups (Fig 5).

In the ECG group, there were fibers, muscles, and connective tissues, but only small amounts of bone tissue, while the amount of bone tissue was higher in the IMG and IFG (Fig 6A). IFG displayed the features of the osteoplastic stage, with homogeneous osteogenic distribution, while IMG displayed inflammatory cell infiltration, a small amount of proliferative connective tissue, and uneven osteogenesis.

Angiography showed that the blood vessels in the ECG group were dissected, and no obvious blood vessels were in the defect area. A small number of new blood vessels were observed in the defect area of the IMG, with continuous but uneven distribution. The quality of the blood vessels formed in the IFG was better, and the local blood supply was more abundant (Fig 6B).

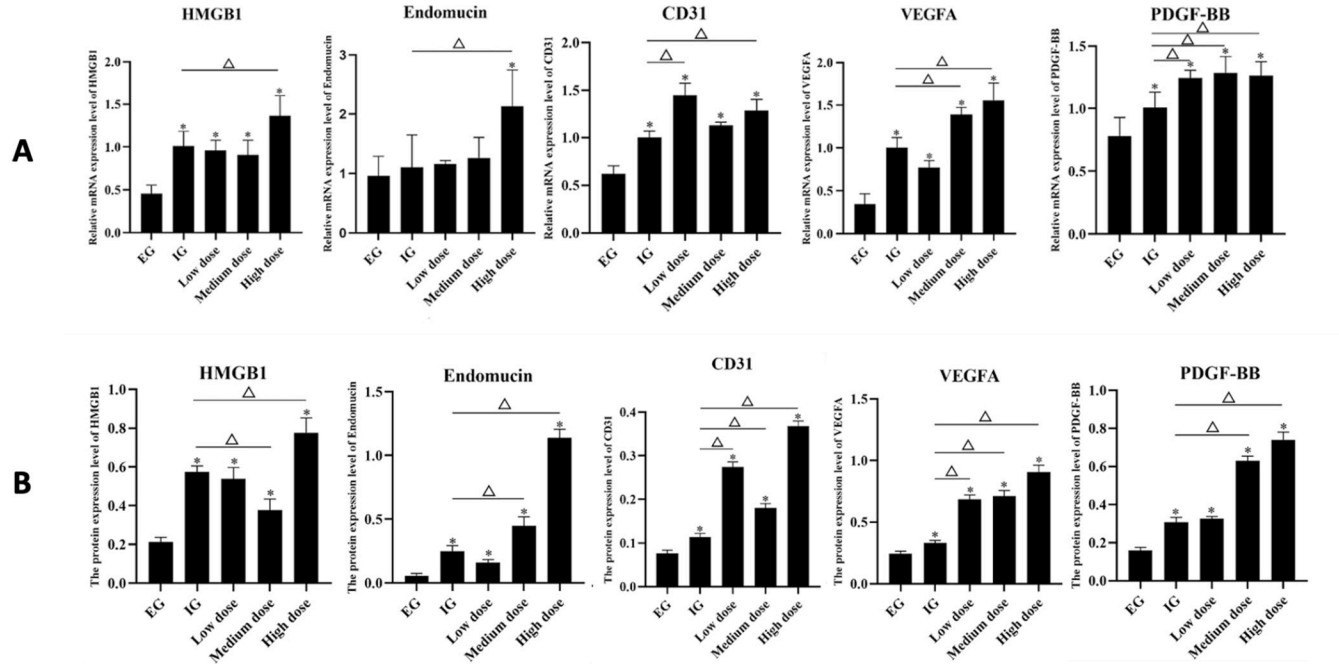

**Fig 3. Expression of genes involved in the ternary regulation of bone defect repair.** (A) mRNA expression (qRT-PCR) of HMGB1, endomucin, CD31, VEGFA, and PDGF-BB in the six groups (empty control group, EG; induced membrane group, IG; low dose RDTF group, LD; medium dose group, MG; high dose group, HG). (B) Protein expression of HMGB1, endomucin, CD31, VEGFA, and PDGF-BB in the six groups $^{\Delta}P<0.05$ according to the line. $^{*}P<0.05$ vs. the EG group.

## 4 Discussion

*Rhizoma drynariae (gusuibu)* is a classic herb used to treat fractures in traditional Chinese medicine [19, 20], the main effective ingredient in *Rhizoma drynariae* is RDTF. CD31$^{hi}$Emcn$^{hi}$ vessels induced by PDGF-BB secreted by osteoclast precursors, together with osteoblasts and osteoclasts, constitute the ternary regulatory mechanism of bone tissue reconstruction [16]. This study aimed to determine whether RDTF can promote bone remodeling and induce membrane growth in the rat Masquelet model and to explore its molecular mechanism based on the ternary regulation theory. The preliminary results showed that RDTF positively affected angiogenesis and bone reconstruction in the bone defect area and on factors involved in ternary regulation.

The formation of the induction membrane is a foreign body reaction of tissues after PMMA implantation, which is embodied by inflammation and edema reaction induced by multinucleated giant cells [27]. At 2 weeks after implantation of PMMA, the induction membrane was observed to have internal and external structures. The inner layer is a synovium-like epithelial cell layer, and the outer layer is mainly composed of fibroblasts, myofibroblasts, and collagen [15, 28]. Masquelet described the induced membrane as a 1 to 2mm thick and has a similar histologically characters to the synovial membrane, providing a microenvironment conducive to bone healing. It was also noted that the induction membrane could provide a very rich blood supply and prevent the soft tissues from growing into the bone defect area and the bone graft absorbed [29–31]. Induced membranes are formed in a complex biological environment in which multiple cytokine systems are involved [32–34]. As reported by Masquelet et al. [8] and Viateau et al. [35, 36], the induced membrane is like periosteum, which both contain and promote regenerative bone cells, growth factors, and neovascularization. Pelissier

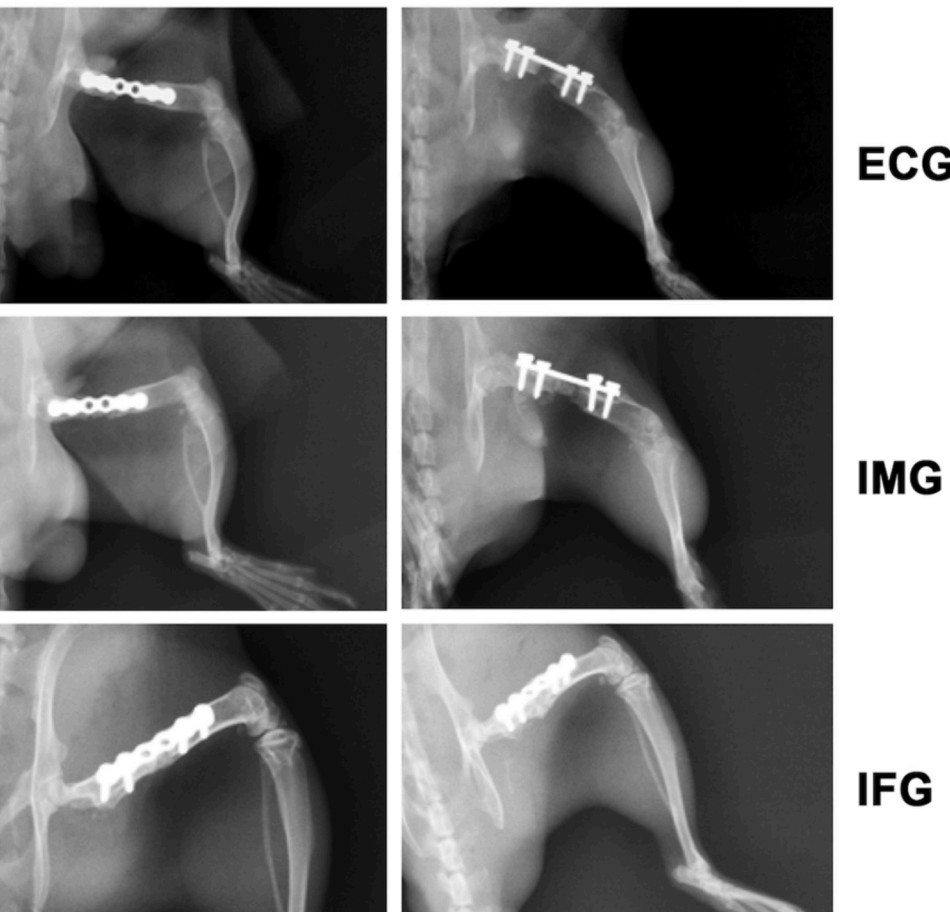

**Fig 4. X-Ray in the bone defect area.** Radiographs of the typical models in the empty control (ECG), induced membrane (IMG), and flavonoid-treated (IFG) groups. For each group, the left image is the anteroposterior view, and the right image is the lateral view.

et al. [9] showed large numbers of capillaries in the outer layer of the induced membrane tissue at various stages, with an average diameter of about 1.2 mm. Tang et al. [37] demonstrated that Dll4/Notch1 signaling is negatively associated with the vessel density of the induced membrane.

In the process of bone repair, the induction membrane acts as a bed for the bone graft and the role of recruiting and chemotactic BMSCs. Wang et al. [38] revealed that the MDSCs of the induced membrane promoted the angiogenesis of endothelial cells through the expression of VEGFA, Ang2, and HIF-1α, which was upregulated by the activation of STAT3 signaling. Due to the affinity between microvessels and osteoprogenitor cells and osteoblasts, the invasion of osteoprogenitor cells into the lesion area in the process of bone repair is often accompanied by vascular growth. There is growing evidence that new blood vessels do more than just establishing local circulation to the new bone and supplying oxygen and nutrients, and they can also directly promote osteogenesis [4]. The expression of neovascularization-related genes in the induced membrane tissue after intervention with RDTF could be used as the basis for future research. This study will also help better understand the biological properties of the induced membrane, which will help create a better microenvironment for bone healing.

This study showed that RDTF had a certain effect on the angiogenesis of the bone defect area, and the number of new vessels in the IFG group was significantly higher than the other

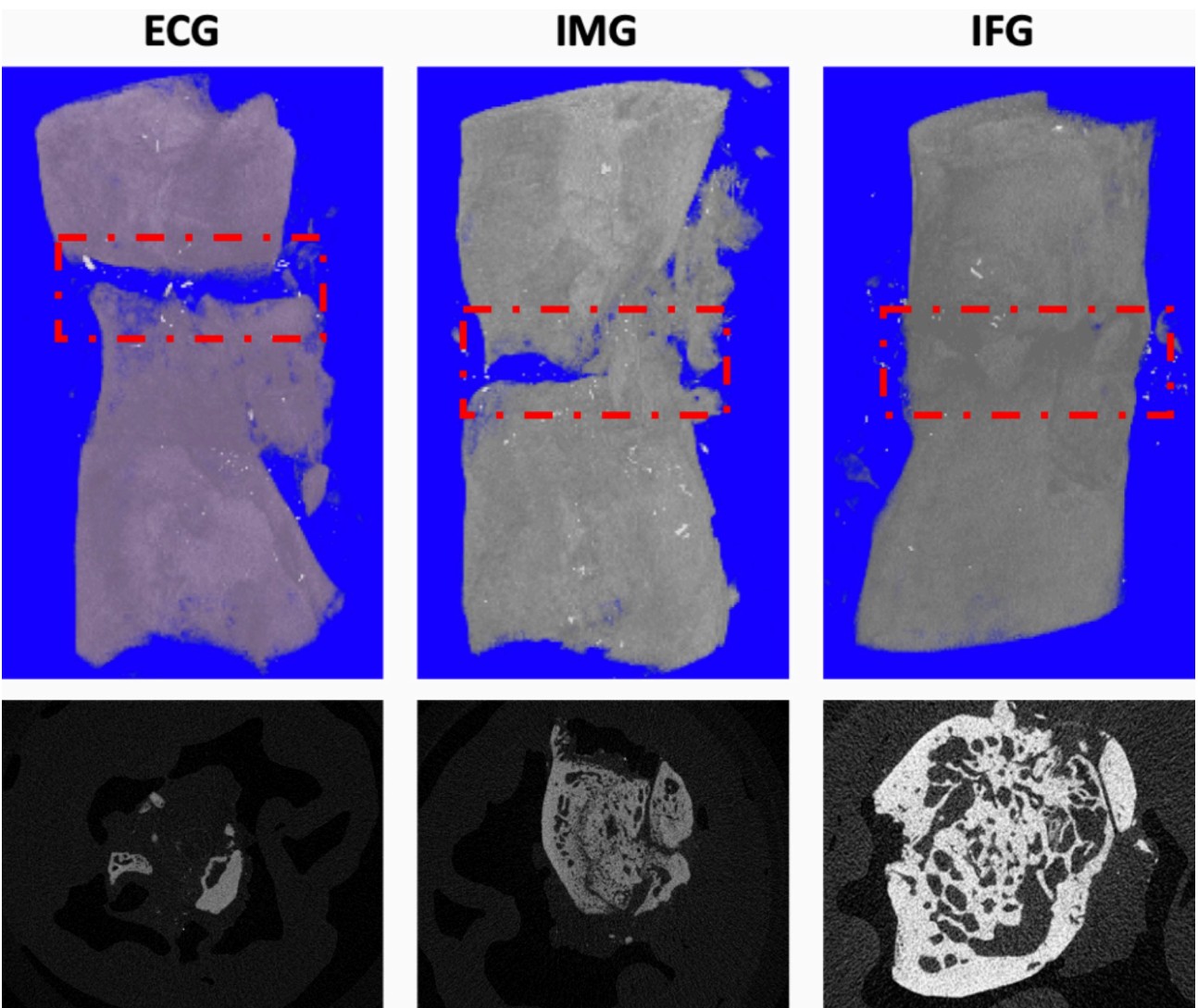

**Fig 5. Micro-computed tomography in the bone defect area.** Micro-CT of the typical models in the empty control (ECG), induced membrane (IMG), and flavonoid-treated (IFG) groups.

groups. X-ray and micro-CT also showed that the mineralization was better than in the other two groups. It could promote the formation of callus in the defect area. From a macroscopical perspective, RDTF in the Masquelet technique positively affects angiogenesis and osteogenesis of the bone defect area.

CD31 expression in endothelial cells is closely related to endothelial cell migration and neo-vascularization and is commonly used as a marker to detect neovascularization [15]. Gouron et al. [15] confirmed that the formation of new blood vessels was continuously maintained during the whole formation process of the inducing membrane. Cuthbert et al. [39] described that CD31 was expressed in the vascular lumen edge in the immunohistochemical results, confirming the existence of abundant blood vessels in the induced membrane tissue. The immunohistochemistry results in this study confirmed the results of Cuthbert et al. [39], and CD31 was shown in the vascular lumen edge of the induced membrane tissue, which was consistent with the results of western blotting and qRT-PCR.

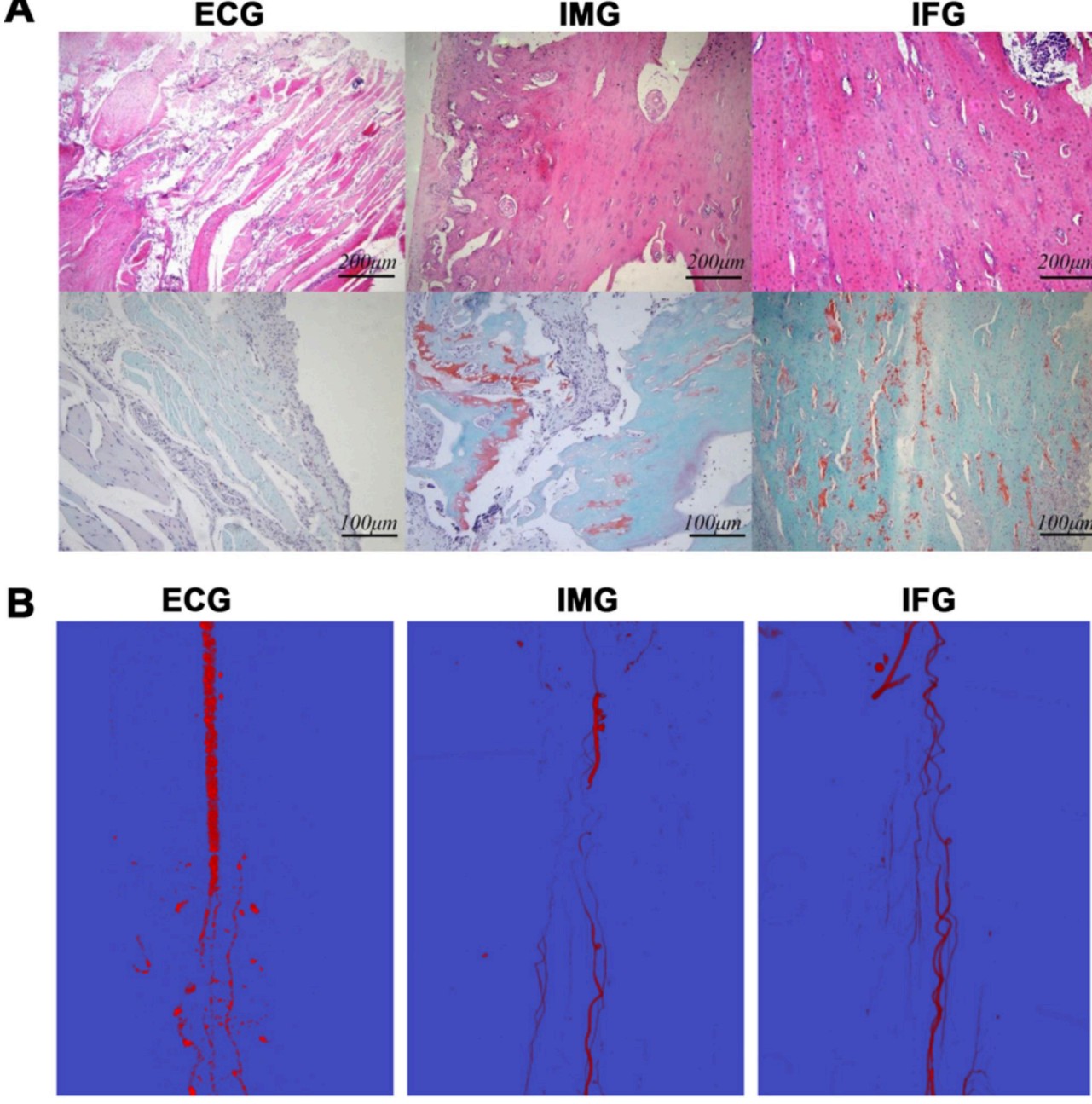

**Fig 6. Effects of RDTF on bone histological examination in the bone defect area.** (A) Hematoxylin and eosin staining (top) and modified red O-solid green cartilage staining (bottom) of the bone defect tissue in the typical models in the empty control (ECG), induced membrane (IMG), and flavonoid-treated (IFG) groups. (B) Angiographic results of bone defect tissue in each group.

Park-Windhol et al. [40] found that endomucin could regulate VEGFR2 activity and affect VEGF-induced endothelial cell migration, growth, and tube morphogenesis. This study suggests that endomucin is a potential regulator of angiogenesis and might provide new targeted therapies for vascular diseases. Different blood vessels could be identified by the immunohistochemistry of specific cells. Using this technique, Kusumbe et al. [41] found that endothelial cells in the endosteum were both strongly positive for CD31 and endomucin, while the venous

sinus located in the diaphysis was only weakly positive for CD31 and weakly expressed endomucin. $CD31^{hi}Emcn^{hi}$ endothelial cells and $CD31^{lo}Emcn^{lo}$ cells were significantly different in the suspension cells of long bone [41]. Therefore, the authors first proposed a new term for intracapsular microvessels: $CD31^{hi}Emcn^{hi}$ vessels, also known as H-type vessels. Endomucin plays an important role in microangiogenesis, and if the immunohistochemistry results show strong CD31 and endomucin expression, the presence of H-type vessels may be suggested. The WB results in the present study confirmed the expression of PDGF-BB protein in the induced membrane tissue, so whether there were H-type blood vessels in the induced membrane tissue was explored. Still, the immunohistochemistry results showed that the expression of endomucin in the induced membrane was weak rather than strong positive, leading to the conclusion that there were no $CD31^{hi}Emcn^{hi}$ vessels in the induced membrane.

This study has limitations. The promoting effects of RDTF on each growth factor were not completely consistent, which might be related to the small sample size of this study and induced membrane sampling operation. The results showed that the concentration of RDTF had a certain effect on the secretion of each growth factor. In this study we didn't performed the experiment in vitro, in the further study we need to do it and to test more bio-markers, in this way we can have a better understanding the molecular mechanism of the ternary regulation theory.

In conclusion, the pathological, X-ray, western blot, and qRT-PCR results confirmed the secretion of VEGF, HMGB1, PDGF-BB, CD31, and endomucin in the induced membrane of the model group was significantly higher than that in the empty control group. Although the expected $CD31^{hi}Emcn^{hi}$ vessels in the induction membrane were not observed, this study confirmed that RDTF could promote the secretion of angiogenic factors in the induced membrane. The specific mechanisms still need to be further studied.

## Supporting information

**S1 Fig. The timeline and design of the experimental and the operation procedure.**
(JPG)

**S2 Fig.** (A) Self-made femoral plate and osteotomy guide plate for rats. (B) The 4-mm precision osteotomy was performed using the osteotomy guide plate to create a rat femoral defect model (panel A). The bone defect area was filled with PMMA cement (panel B). Four weeks after the primary operation, the induction of the bone membrane was observed (panel C). The induced membrane tissue was taken out (panel D).
(JPG)

## Author Contributions

**Conceptualization:** Ding Li, Yue Li.

**Data curation:** Ding Li, Dun Zhao, Zhikui Zeng.

**Formal analysis:** Feng Huang.

**Funding acquisition:** Bin Fang, Yue Li.

**Investigation:** Hao Xiong, Tianan Guan.

**Methodology:** Ding Li, Zhikui Zeng, Yue Li.

**Project administration:** Ziwei Jiang.

**Resources:** Ding Li, Dun Zhao.

**Software:** Ding Li, Dun Zhao.

**Supervision:** Feng Huang, Yue Li.

**Validation:** Ziwei Jiang, Bin Fang.

**Visualization:** Ding Li.

**Writing – original draft:** Ding Li, Dun Zhao, Zhikui Zeng.

**Writing – review & editing:** Yue Li.

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
