## [Decision Letter · Decision Letter 0]

12 Aug 2022

PONE-D-22-18003Ternary regulation mechanism of Rhizoma drynariae total flavonoids on induced membrane formation and bone remodeling in Masquelet techniquePLOS ONE

Dear Dr. Li,

Thank you for submitting your manuscript to PLOS ONE. After careful consideration, we feel that it has merit but does not fully meet PLOS ONE’s publication criteria as it currently stands. Therefore, we invite you to submit a revised version of the manuscript that addresses the points raised during the review process.

We look forward to receiving your revised manuscript.

Kind regards,

Selvaraj Vimalraj

Academic Editor

PLOS ONE

Journal Requirements:

"The authors report no conflict of interest."

Additional Editor Comments:

Carefully consider the reviewers' comments and suggested to submit a revised manuscript.

Reviewers' comments:

Reviewer's Responses to Questions

**Comments to the Author**

1. Is the manuscript technically sound, and do the data support the conclusions?

Reviewer #1: Yes

Reviewer #2: Yes

2. Has the statistical analysis been performed appropriately and rigorously? 

Reviewer #1: Yes

Reviewer #2: Yes

3. Have the authors made all data underlying the findings in their manuscript fully available?

Reviewer #1: Yes

Reviewer #2: Yes

4. Is the manuscript presented in an intelligible fashion and written in standard English?

Reviewer #1: Yes

Reviewer #2: Yes

5. Review Comments to the Author

Reviewer #1: Comments

1. What is the rationale behind choosing oral administration of RDTF rather than other modes of drug administration or even any implantation modes?

2. Did the author carry out experiments on the study of in vitro cyto-compatibility of the drug in any means?

3. Did the author go through literatures having similar work performed in any other animal models?

4. Kindly check the spelling for Histological in page 16 (Line 188), Quantitative Page 17 (Line 196), Page 18 (Line 217), Page 18 (Line 236) and change it wherever applicable.

5. Justify the reason for selecting these particular concentrations of the drug for animal experiment? Why the author did not try with series of concentrations of the drug but ended up with one particular concentration for HG, MG, LG in animal study? Please justify.

6. The author has performed experiments on animal model to evaluate the levels of proteins and other biomarkers but failed to work on the extraction of individual bio-active flavonoid and its role in protein expression at in vitro cellular level. How would you validate that the present study with whole tuber flavonoid extract of drynaria gives enough stuff on ternary regulation of bone defect in animal model?

7. It is validated that Rhizoma drynaria contains HCN and did the author work on any procedures to remove it before extracting the total flavonoids.

8. Also, did the author check the anti-oxidant activity of the compound using in vitro biological assays specially to study the effect of oxidative stress markers?

9. In page 10, (line 64-66) the sentence “After 4 to 8 weeks a membrane surrounding the spacer was formation, then the second stage surgery was operated, including removal of the spacer and implant the bone grafts (citation).” can be changed to “After 4 to 8 weeks, a membrane surrounding the spacer was formed and then the second stage surgery was operated to remove the spacer for implantation of bone grafts (Include citation) “

10. The author has claimed that the autografts implanted are rat coccyx. Justify how did the author manages post-operative pain for the animal, risk of fracture at the donor site.

11. Is there any quantified data on mesenchymal stem cell (MSCs) production in induced membranes of IFGs/ECG/IMG with respect to normal bone marrow of their respective groups to validate the osteo-angiogenic potential

12. Did the author perform any experiments with induction membranes retrieved from reconstructed CSD femur model to quantify angiogenic and osteogenic bio-markers

13. Did the author validate any results on soft callus regeneration during bone remodeling with PMMA like how it is done after bone graft repair? Since this method includes long treatment period, did the author experimented on early bone healing phases during PMMA post-surgery?

14. The author has claimed that the induction membrane serves as periosteum like matrix to help in cell proliferation, growth factor production and signaling but the expected study did not prove the pro-angiogenic ability of IMGs individually. The gene and protein expression studies as mentioned in Fig 3A and 3B for VEGFA and CD31 showed minimal effect. Justify why PECAM and VEGFA expressions are hindered in IMGs than other groups.

Reviewer #2: The manuscript titled on " Ternary regulation mechanism of Rhizoma drynariae total flavonoids on inducedmembrane formation and bone remodeling in Masquelet technique" is an interesting work done by author. However, there are few questions need to be addressed in this study.

1) what is the rationale for choosing Rhizoma drynariaetotal flavonoids(RDTF) ? There are several flavonoids currently used for treating bone defect. How different from other study ?

2) Bone remodeling involves the secretion of bone markers such as ALP, OC etc. Do an author checked the expression of these other than VEGF and PDGF?

3) section 2.4-2.7 contains lot of spelling mistake in the title. For example, instead of histological author wrote ological analysys. It needs to be corrected prior to publication.

Fig.4 legends are not visible, poor resolution. Need a better resolution. In the RT-PCR, author didn't explain how did normalization done. And what method did you use to calculate mRNA expresssion ?

Do an author checked any of the protein expression such as VEGF and PDGF?

6. PLOS authors have the option to publish the peer review history of their article (what does this mean?). If published, this will include your full peer review and any attached files.

Reviewer #1: **Yes: **DR. DHARUNYA GOVINDARAJAN

Reviewer #2: **Yes: **Arumugam Balasubramanian

---

## [Author Response · Author response to Decision Letter 0]

30 Oct 2022

Response to Reviewers 1

1.What is the rationale behind choosing oral administration of RDTF rather than other modes of drug administration or even any implantation modes?

RDTF is extracted from a traditional Chinese herb Rhizoma drynariae (Chinese name, Gusuibu), and it is dissolved well in water can be well absorbed by the gastrointestinal system, so we choose oral administration of RDTF rather than other modes of drug administration.

2. Did the author carry out experiments on the study of in vitro cyto-compatibility of the drug in any means?

Our groups have done some experiments on the study of in vitro cyto-compatibility of the drug

3. Did the author go through literatures having similar work performed in any other animal models?

Yes, some studies are focused on the osteoporosis models.

4. Kindly check the spelling for Histological in page 16 (Line 188), Quantitative Page 17 (Line 196), Page 18 (Line 217), Page 18 (Line 236) and change it wherever applicable.

Thanks for the remind of the details, we will revise them carefully.

5. Justify the reason for selecting these particular concentrations of the drug for animal experiment? Why the author did not try with series of concentrations of the drug but ended up with one particular concentration for HG, MG, LG in animal study? Please justify.

The dose is designed according to the clinical effective dose, and the HG, MG, LG is according the human effective dosage, and conversion of human and animal administration.

6. The author has performed experiments on animal model to evaluate the levels of proteins and other biomarkers but failed to work on the extraction of individual bio-active flavonoid and its role in protein expression at in vitro cellular level. How would you validate that the present study with whole tuber flavonoid extract of drynaria gives enough stuff on ternary regulation of bone defect in animal model?

In this study we just performed on the animal model, and do not perform in vitro. In the future study we will test in vitro cellular level.

7. It is validated that Rhizoma drynaria contains HCN and did the author work on any procedures to remove it before extracting the total flavonoids.

The total flavonoids are purchase from the company; we didn’t remove the HCN by ourselves.

8. Also, did the author check the anti-oxidant activity of the compound using in vitro biological assays specially to study the effect of oxidative stress markers?

This is a very good suggestion for the next step study. Anti-oxidant is also very important in the vessel formation, in this study we didn’t test the oxidative stress markers, but in the future study we can do it.

9. In page 10, (line 64-66) the sentence “After 4 to 8 weeks a membrane surrounding the spacer was formation, then the second stage surgery was operated, including removal of the spacer and implant the bone grafts (citation).” can be changed to “After 4 to 8 weeks, a membrane surrounding the spacer was formed and then the second stage surgery was operated to remove the spacer for implantation of bone grafts (Include citation) “

Thanks for the remind, will change it.

10. The author has claimed that the autografts implanted are rat coccyx. Justify how did the author manages post-operative pain for the animal, risk of fracture at the donor site.

We use the Meloxicam（0.2mg/kg）in the first 3 days after operative, carefully separate the tissand suture the donor site , no animals found fracture in this study.

11. Is there any quantified data on mesenchymal stem cell (MSCs) production in induced membranes of IFGs/ECG/IMG with respect to normal bone marrow of their respective groups to validate the osteo-angiogenic potential. 

In this study we didn’t test this, in future study we can test it.

12. Did the author perform any experiments with induction membranes retrieved from reconstructed CSD femur model to quantify angiogenic and osteogenic bio-markers

Yes. Our group have done some experiments to test the induction membranes angiogenic and osteogenic bio-markers.

13. Did the author validate any results on soft callus regeneration during bone remodeling with PMMA like how it is done after bone graft repair? Since this method includes long treatment period, did the author experimented on early bone healing phases during PMMA post-surgery?

This is a very important concern of the bone regeneration during the PMMA, in this study we didn’t do this, but it can be revolved in the further study.

14. The author has claimed that the induction membrane serves as periosteum like matrix to help in cell proliferation, growth factor production and signaling but the expected study did not prove the pro-angiogenic ability of IMGs individually. The gene and protein expression studies as mentioned in Fig 3A and 3B for VEGFA and CD31 showed minimal effect. Justify why PECAM and VEGFA expressions are hindered in IMGs than other groups.

The results showed in Fig 3A and 3B demonstrate that compare to empty control group (EG), the induced membrane group (IG) has higher level in CD31 and VEGFA, this suggest that the induced membrane may has a pro-angiogenic ability compare to control group. Because the RDTF has the effect on angiogenic, so in groups of the LG、MG、HG has a higher level of VEGFA and CD31.

Response to Reviewers 2

1) what is the rationale for choosing Rhizoma drynariaetotal flavonoids(RDTF) ? There are several flavonoids currently used for treating bone defect. How different from other study ?

2) Bone remodeling involves the secretion of bone markers such as ALP, OC etc. Do an author checked the expression of these other than VEGF and PDGF?

Yes, in this study we didn’t test the ALP, OC, in this study we focused on the biomarkers of the vessel’s and in future study we will test the relationship between the bone markers and the vessels biomarkers, this is a very important suggestion of the further study.

3) section 2.4-2.7 contains lot of spelling mistake in the title. For example, instead of histological author wrote ological analysys. It needs to be corrected prior to publication.

Fig.4 legends are not visible, poor resolution. Need a better resolution. In the RT-PCR, author didn't explain how did normalization done. And what method did you use to calculate mRNA expresssion ?Do an author checked any of the protein expression such as VEGF and PDGF?

Thanks for the remind, we will revise the details.

Thanks for the kindly remind of the details, we will revise it.

For the normalization and calculate has add in the methods, The expression of each gene was normalized to GAPDH expression by using the 2-ΔΔCt method.

In this study we just use the RT-PCR and Western-blot to test the biomarkers.

---

## [Decision Letter · Decision Letter 1]

22 Nov 2022

Ternary regulation mechanism of Rhizoma drynariae total flavonoids on induced membrane formation and bone remodeling in Masquelet technique

PONE-D-22-18003R1

Dear Dr. Li,

We’re pleased to inform you that your manuscript has been judged scientifically suitable for publication and will be formally accepted for publication once it meets all outstanding technical requirements.

Kind regards,

Selvaraj Vimalraj

Academic Editor

PLOS ONE

Additional Editor Comments (optional):

accept

Reviewers' comments:

Reviewer's Responses to Questions

**Comments to the Author**

1. If the authors have adequately addressed your comments raised in a previous round of review and you feel that this manuscript is now acceptable for publication, you may indicate that here to bypass the “Comments to the Author” section, enter your conflict of interest statement in the “Confidential to Editor” section, and submit your "Accept" recommendation.

Reviewer #1: All comments have been addressed

Reviewer #2: All comments have been addressed

2. Is the manuscript technically sound, and do the data support the conclusions?

Reviewer #1: Partly

Reviewer #2: Yes

3. Has the statistical analysis been performed appropriately and rigorously? 

Reviewer #1: Yes

Reviewer #2: N/A

4. Have the authors made all data underlying the findings in their manuscript fully available?

Reviewer #1: Yes

Reviewer #2: Yes

5. Is the manuscript presented in an intelligible fashion and written in standard English?

Reviewer #1: Yes

Reviewer #2: Yes

6. Review Comments to the Author

Reviewer #1: Kindly check the spelling and grammatical error and make changes wherever applicable. The answers are satisfying and so the manuscript can be forwarded for publication.

All the best.

Reviewer #2: The author clearly addressed all the comments raised in this manuscript. Therefore the manuscript can be accepted for publication.

7. PLOS authors have the option to publish the peer review history of their article (what does this mean?). If published, this will include your full peer review and any attached files.

Reviewer #1: **Yes: **DR. DHARUNYA GOVINDARAJAN

Reviewer #2: No

---

## [Editor Report · Acceptance letter]

24 Nov 2022

PONE-D-22-18003R1 

Ternary regulation mechanism of *Rhizoma drynariae*  total flavonoids on induced membrane formation and bone remodeling in Masquelet technique 

Dear Dr. Li:

I'm pleased to inform you that your manuscript has been deemed suitable for publication in PLOS ONE. Congratulations! Your manuscript is now with our production department. 

Kind regards, 

on behalf of

Dr. Selvaraj Vimalraj 

Academic Editor

PLOS ONE